# High mitochondrial diversity of domesticated goats persisted among Bronze and Iron Age pastoralists in the Inner Asian Mountain Corridor

Taylor R. Hermes[1,2¤]*, Michael D. Frachetti[3], Dmitriy Voyakin[4,5], Antonina S. Yerlomaeva[6], Arman Z. Beisenov[6], Paula N. Doumani Dupuy[7], Dmitry V. Papin[8,9], Giedre Motuzaite Matuzeviciute[10], Jamsranjav Bayarsaikhan[11], Jean-Luc Houle[12], Alexey A. Tishkin[13], Almut Nebel[14], Ben Krause-Kyora[14], Cheryl A. Makarewicz[1,2]*

1 Graduate School "Human Development in Landscapes", Kiel University, Kiel, Germany, 2 Institute of Prehistoric and Protohistoric Archaeology, Kiel University, Kiel, Germany, 3 Department of Anthropology, Washington University in St. Louis, St. Louis, Missouri, United States of America, 4 Archaeological Expertise, LLC, Almaty, Kazakhstan, 5 International Institute for Central Asian Studies, Samarkand, Uzbekistan, 6 Margulan Institute of Archaeology, Almaty, Kazakhstan, 7 School of Humanities and Social Sciences, Nazarbayev University, Nur-Sultan, Kazakhstan, 8 The Laboratory of Interdisciplinary Studies in Archaeology of Western Siberia and Altai, Altai State University, Barnaul, Russia, 9 Institute of Archaeology and Ethnography, Siberian Branch of the Russian Academy of Sciences, Novosibirsk, Russia, 10 Department of Archaeology, Vilnius University, Vilnius, Lithuania, 11 National Museum of Mongolia, Ulaanbaatar, Mongolia, 12 Department of Folk Studies and Anthropology, Western Kentucky University, Bowling Green, Kentucky, United States of America, 13 Department of Archaeology, Ethnography and Museology, Altai State University, Barnaul, Russia, 14 Institute of Clinical Molecular Biology, Kiel University, University Hospital Schleswig-Holstein, Kiel, Germany

¤ Current address: Department of Archaeogenetics, Max Planck Institute for the Science of Human History, Jena, Germany

* hermes@shh.mpg.de (TRH); c.makaewicz@ufg.uni-kiel.de (CAM)

**Data Availability Statement:** All relevant data are within the manuscript and its Supporting Information files.

## Abstract

Goats were initially managed in the Near East approximately 10,000 years ago and spread across Eurasia as economically productive and environmentally resilient herd animals. While the geographic origins of domesticated goats (*Capra hircus*) in the Near East have been long-established in the zooarchaeological record and, more recently, further revealed in ancient genomes, the precise pathways by which goats spread across Asia during the early Bronze Age (ca. 3000 to 2500 cal BC) and later remain unclear. We analyzed sequences of hypervariable region 1 and cytochrome *b* gene in the mitochondrial genome (mtDNA) of goats from archaeological sites along two proposed transmission pathways as well as geographically intermediary sites. Unexpectedly high genetic diversity was present in the Inner Asian Mountain Corridor (IAMC), indicated by mtDNA haplotypes representing common A lineages and rarer C and D lineages. High mtDNA diversity was also present in central Kazakhstan, while only mtDNA haplotypes of lineage A were observed from sites in the Northern Eurasian Steppe (NES). These findings suggest that herding communities living in montane ecosystems were drawing from genetically diverse goat populations, likely sourced from communities in the Iranian Plateau, that were sustained by repeated interaction and exchange. Notably, the mitochondrial genetic diversity associated with goats of the

**Funding:** This research was supported by the doctoral fellowship of T.R.H. at the Graduate School "Human Development in Landscapes" (German Research Foundation: GSC 208). This project has received funding from the European Research Council (ERC) under the European Union's Horizon 2020 research and innovation program (ERC Consolidator Grant-772957/ ASIAPAST held by C.A.M.). Archaeological research at Tasbas and Dali was funded by Washington University in St. Louis and the United States National Science Foundation (no. 1132090 held by P.N.D.D and M.D.F.). Archaeological research at Begash was funded by United States National Science Foundation (nos. 0211431 and 0535341 held by M.D.F.). Archaeological research at Uch-Kurbu was funded by the European Social Fund according to the activity "Improvement of researchers" qualification by implementing world-class R&D projects' of Measure (no. 09.3.3-LMT-K-712 held by G.M.M.). Archaeological research at Zamiin-Utug was funded by Western Kentucky University (J-L.H.). This research was also supported by the Russian Science Foundation, project no. 16-18-10033 "Formation and Evolution of the Subsistence System of the Nomadic Societies of Altai and Adjacent Territories in Late Antiquity and the Middle Ages: Complex Reconstruction" held by A.A.T. Archaeological Expertise, LLC provided support in the form of salaries for author D.V., but did not have any additional role in the study design, data collection and analysis, decision to publish, or preparation of the manuscript. The specific roles of these authors are articulated in the 'author contributions' section.

**Competing interests:** Author DV is affiliated with Archaeological Expertise, LLC, but this does not alter our adherence to PLOS ONE policies on sharing data and materials. There are no patents or products in development at Archaeological Expertise, LLC that are relevant to this work

IAMC also extended into the semi-arid region of central Kazakhstan, while NES communities had goats reflecting an isolated founder population, possibly sourced via eastern Europe or the Caucasus region.

## Introduction

Domesticated goats (*Capra hircus*) are at the subsistence core of pastoralist and farming communities around the world, providing dependable sources of meat, milk, fiber and hides [1]. Goats are remarkably resilient in challenging environments, having a broader foraging spectrum and requiring less water than other livestock taxa [2–4]. Wild bezoar (*Capra aegagrus*) goats were under initial management in various regions of the Near East by the mid-ninth millennium BC [5, 6]. By the eighth millennium BC, goats were closely controlled via husbandry practices that adjusted animal diets, movement, and herd demography favoring female survivorship [7–11], and also exhibited considerable size diminution and loss of wild phenotypic traits associated with the domestication syndrome, indicating that people had effectively reconfigured the fitness landscape of managed goats and genetically isolated them from sympatric bezoars [12, 13].

Recent paleogenetic research [14] and previous studies [15–17] established that goat domestication processes involved multiple maternal origins throughout the Near East and Iranian Plateau, giving rise to early phylogeographic structure of six divergent mtDNA haplogroups (A, B, C, D, F, and G). However, post-Neolithic domesticated goat populations across these regions show a collapse of this initial mtDNA phylogeographic pattern, leading to widespread dominance of A lineages [14], which today characterize 91% of domesticated goats world-wide [15–19]. Crucially, as Neolithic technologies and possibly people spread from centers of plant and animal domestication, mtDNA phylogeographic patterns may have been transmitted to adjacent and distant regions across Eurasia, thus, revealing goat transmission routes. Moreover, goat herds of later Bronze and Iron Age peoples might have been characterized by shifts in the relative abundance of mtDNA haplotypes on account of the geographic extent and intensity of social interaction spheres in which goats were bred and exchanged.

Central Asia is a critical region for examining the spread of domesticated plants and animals across the vast expanse of Eurasia, which led to widespread transformations in human subsistence and socio-political structures [20]. In Central Asia, regionally varied pastoralist lifeways developed in continental environments much different from the diverse landscapes of the Near East, where the earliest forms of mobile pastoralism appears to have first emerged in the late seventh-early sixth millennium BC, when specialized herders began residing in semi-permanent settlements located in the more marginal semi-arid eastern steppes of the Levant and possibly practiced residential mobility on a seasonal basis [21–24]. There are two hypothesized routes for the transmission of subsistence herding into Central Asia and farther afield. The first is the "Northern Eurasian Steppe" (NES), a distinct ecotone of grasslands at roughly 50–55°N latitude from eastern Europe to southern Siberia (Fig 1) [25]. Archaeologists have long thought that diverse herding communities from the Pontic-Caspian steppe known as the "Yamnaya" engaged in a series of eastward migrations along the NES to the Altai Mountains and then south into Central Asia (ca. 3300–2500 BC), which coincides with pronounced changes in material culture [26], such as a distinct mortuary practice of stone enclosures and ceramic styles resembling that of contemporaneous Yamnaya sites [27–29]. Critically, human archaeogenetic research documents close genetic links between these so-called "Afanasievo"

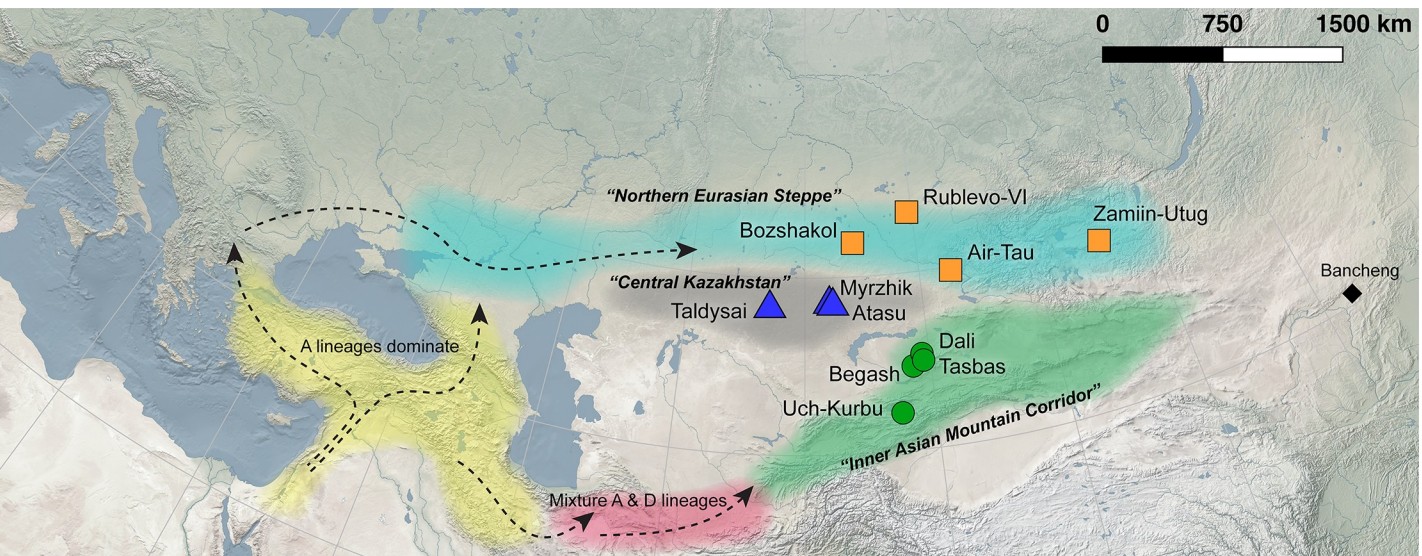

**Fig 1. Map of Eurasia showing sites analyzed in this study in colorful shapes corresponding to analytical regions.** Post-Neolithic distribution of mitochondrial lineages across the Middle East follows Daly et al. [14], and supposed dispersal routes of domesticated goats are shown with dashed arrows. Made with Natural Earth. Free vector and raster map data @ http://www.naturalearthdata.com.

and "Yamnaya" groups [30–33]. In later periods, the NES arguably functioned as a cultural exchange thoroughfare over which key steppe innovations were transmitted [34], such as bronze technologies [35, 36] and domesticated horses [26, 37, 38]. By the Iron Age, the NES represents as a critical arena for the geographically expansive Iron Age "Scythian" cultural community (ca. 500 cal BC) and subsequent growth of "nomadic empires" [39–44].

A second route for the spread of pastoralist subsistence to Central Asia reflects a vector of cultural transmission from the Iranian Plateau via a near continuous series of foothill ecozones to the Altai Mountains, dubbed the "Inner Asian Mountain Corridor" (IAMC) (Fig 1) [20]. Accordingly, pastoralist subsistence (perhaps through population movement or cultural transmission) is thought to have spread into Central Asia during the fourth millennium BC along montane elevational gradients as a strategy to provide graze for livestock by exploiting seasonal rhythms of vegetation growth [45]. Recent archaeological research in the IAMC has documented pastoralist occupations dating to the early third millennium BC that coincided with the initial trans-Eurasian transmissions of domesticated cereals of wheat and barley from the Near East and broomcorn millet from northern China [46–48]. Human paleogenetic research suggests that the IAMC in the second millennium BC was a distinct conduit for the northward human gene flow from agro-pastoral communities from the region of Turan, roughly spanning present-day Turkmenistan and Uzbekistan [31]. In subsequent millennia, pastoralist interactions throughout the IAMC are associated with vibrant cultural exchanges, linking economies and cultures across Eurasia, which scholars commonly refer to as the ancient Silk Roads [45, 49–54].

While the NES and IAMC served as major cultural interaction spheres over the past six thousand years, there has been very little systematic evaluation of the dispersals of pastoralist subsistence or animal exchanges within and between these geographic loci. Here, we examine the mitochondrial (mtDNA) diversity of domesticated goats from sites located in the NES and IAMC with Bronze Age and/or Iron Age occupations in order to reconstruct the spread of this key pastoralist animal to Central Asia and subsequent genetic patterns (Fig 1) [55]. Key Bronze Age sites located between the NES and IAMC provide a test for the geographic extent of

maternally inherited goat genetic variation potentially characteristic of these macro regions (Fig 1). We analyzed the hypervariable region 1 (HVR1) of the control region and cytochrome *b* gene (*MT-CYB*) recovered from ancient goat skeletal remains. HVR1 sequences contain high frequencies of polymorphisms that characterize particular haplogroups, while *MT-CYB* sequences are highly conserved and generally reflect inter-species variation, which we use here to differentiate between domesticated sheep and goat for ambiguously identified archaeological specimens, in addition to wild *Capra* species [56], which are likely present in the zooarchaeological assemblages of the sampled sites. Prior to a major study across the Near East [14], a very limited number of ancient mtDNA sequences have been analyzed, which only represent Neolithic Europe [57] and the Near East [58–60] and also Iron Age China [61, 62].

## Materials and methods

### Sample selection, preparation, and DNA extraction

All necessary permits were obtained for the described study, which complied with all relevant regulations. Bones and teeth from 198 probable or possible goats, identified by morphological features [63, 64] were selected from 11 archaeological sites with Bronze and/or Iron Age occupations in Kazakhstan, Kyrgyzstan, Russia, and Mongolia (Fig 1; Table 1, S1 Table). Wear stages of mandibular premolars and molars were scored following Payne [65], in order to differentiate individuals in case of recovering identical haplotypes from multiple specimens at each site. Laboratory work was conducted in the Ancient DNA Laboratory of Kiel University following established protocols [66–69]. Negative controls were employed during each set of DNA extractions and PCR preparations.

**Table 1. Analytical regions and sites used in this study.** Archaeological descriptions of sites are available in S1 Text.

| Region/Site | Chronology | Country | Elevation (m) | Description | Reference |
|---|---|---|---|---|---|
| Northern Eurasian Steppe | | | | | |
| Bozshakol | Late Bronze Age | Kazakhstan | 225 | Metallurgical complex occupied from 1600 to 1400 cal BC. | [76] |
| Air-Tau | Late Bronze Age | Kazakhstan | 260 | Settlement site occupied during the first half of the second millennium BC, based on ceramic typologies. | Unpublished |
| Rublevo-VI | Late Bronze Age | Russia | 160 | Settlement site occupied from 1400 to 1000 cal BC. | [77–80] |
| Zamiin-Utug | Iron Age | Mongolia | 1800 | Xiongnu cemetery site used from AD 200 to 100. | Unpublished |
| Central Kazakhstan | | | | | |
| Taldysai | Middle-Late Bronze Age | Kazakhstan | 470 | Metallurgical complex occupied from 1900 to 1200 cal BC. | [81] |
| Myrzhik | Late Bronze Age | Kazakhstan | 500 | Settlement site occupied during the second half of the second millennium BC, based on ceramic typologies. | [76, 81–84] |
| Atasu | Late Bronze Age | Kazakhstan | 480 | Settlement site occupied during the second half of the second millennium BC, based on ceramic typologies. | [81, 84] |
| Inner Asian Mountain Corridor | | | | | |
| Begash | Early Bronze Age to modern | Kazakhstan | 800 | Settlement site occupied from 2400 cal BC to the early 20th c. | [47, 48, 85–90] |
| Dali | Early-Late Bronze Age | Kazakhstan | 1500 | Settlement site occupied from 2700 to 1100 cal BC. | [47, 89] |
| Tasbas | Late Bronze Age | Kazakhstan | 1500 | Settlement site occupied from 2800 to 800 cal BC. | [46, 47, 87, 89–91] |
| Uch-Kurbu | Late Bronze Age | Kyrgyzstan | 1750 | Cemetery site used from 1700 to 1300 cal BC. | [92] |

Samples were soaked in consumer bleach solution for 5 min in order to remove external contamination, followed by a soaking for 5 min and two rinses with ddH$_2$O. Samples were ground to a fine powder with a ball mill and digested in 960 μL 0.45 M EDTA (pH 8.0) and 40 μL proteinase K (25 mg/mL) overnight at 37˚C. DNA was extracted with a large-volume extraction protocol using a guanidinium-silica based method [70]. Two independent DNA extractions from sample powder were performed for each specimen yielding an *MT-CYB* sequence of *Capra hircus*, and sample amplicons were each sequenced twice, in forward and reverse directions.

## mtDNA sequence analysis

A 110-bp fragment in the cytochrome *b* gene for taxonomic identification among *Ovis* and *Capra* spp. was targeted using primers CapFC1 (5'– CTCTGTAACTCACATTTGTC–3') and CapRB1b (5'– GTTTCATGTTTCTAGAAAGGT–3') [71]. This initial genetic identification of species is crucial, especially since Central Asia is home to wild sheep and goats, including argali (*Ovis ammon*), urial (*Ovis vignei*), Siberian ibex (*Capra sibirica*), and markhor (*Capra falconeri*), which may be misidentified and confused with domesticated sheep and goat if relying on skeletal morphological traits. Primers CAP-FII (5'– GATCTTCCYCATGCATATAA GCA–3') and CAP-RIII (5'– GATAAAGTTCATTAAATAGCTAC–3') targeted a 193-bp fragment in the control region of *Capra* spp. containing diagnostic polymorphisms for determining haplogroups and haplotypes [60]. In case the 193-bp amplification failed, a shorter 130-bp fragment of the control region was targeted using primers CAP-FII and CAP-RII (5'– CGGGTTGCTGGTTTCAC–3') [71]. Finally, while primer pairs that were used for domesticated goats were also used for *Capra sibirica* specimens, an additional pair of primers, CSib-F1 (5'– AACATGCGTATCCCGTCCAC–3') and CSib-R1 (5'– GGCCCTGAAGAAAGAACC AG–3'), were designed for amplifying a 120-bp fragment of the mtDNA control region overlapping with the previously generated 193-bp amplicon, resulting in a final control region sequence of 221-bp. This allowed for better comparison with a limited number of modern *Capra sibirica* mtDNA sequences.

Amplifications were performed as singleplex PCRs in a final volume of 25 μL (S2 Table). Reactions were performed in a Mastercycler (Eppendorf) with the following conditions: initialization at 94˚C for 10 min, 42 cycles of 94˚C for 30 s, 50–60˚C for 30 s (primer dependent, see S3 Table), 72˚C for 30 s, and a final elongation step at 72˚C for 10 min. The quality of PCR products was checked on a QIAxcel system, and those with single bands of expected size were subjected to direct sequencing with the BigDye Terminator v3.1 Kit (Applied Biosystems), following the instructions of the manufacturer. Sequencing products were purified applying the DyeEx 2.0 Spin Kit (Qiagen) and analyzed using the ABI Prism 310 and 3130 Genetic Analyzers (Applied Biosystems).

Ancient *Capra hircus* sequences of the control region and cytochrome *b* gene were concatenated for each sample, resulting in a total sequence length of 303-bp. Sequences were pair-wise aligned to one another in addition to nine previously published ancient sequences and 10 modern sequences of *Capra* spp. representing haplogroups A, B, C, D, F, and G (S4 Table). In cases where reference sequences of the control region did not accompany sequences of cytochrome *b*, we used cytochrome *b* sequences from sample KG170121. This approach is justified because of very low genetic variation in cytochrome *b* sequences for each *Capra* sp., which mainly served to improve phylogenetic clustering of wild Siberian ibex relative to domesticated goats. Sequences were trimmed to exclude the primer binding sites. A phylogenetic tree was constructed using MrBayes v. 3.2.6 [72] in Geneious v. 10.0.7. The simulation was set using the JC69 substitution model, invgamma rate variation, four gamma categories,

and a sequence from *Ovis aries* as an outgroup; MCMC chain length was set to 500,000 with a sampling frequency of 200 and burn-in of 100,000. Additionally, a minimum-spanning haplotype network of sample and reference sequences was generated using PopART v. 1.7 [73]. Nucleotide diversity was calculated for each analytical region from the number of base substitutions per nucleotide position from averaging over all sequence pairs using the Kimura 2-parameter model and gamma distribution shape parameter of 1 [74], with standard error estimates by a bootstrap procedure with 1000 replicates in MEGA X v.10.1.1 [75]. Only sequences represented by 303-bp without gaps were included in the genetic distance analysis.

## Results and discussion

### *MT-CYB* sequence recovery and taxonomic identification

Out of 198 ancient skeletal specimens identified as probable goats or ambiguously identified as sheep/goat (caprine), we recovered 50 *MT-CYB* sequences of *Capra hircus*, 94 *MT-CYB* sequences of *Ovis aries*, and two *MT-CYB* sequences of *Capra sibirica* (Table 2). The relatively high overall recovery rate of 73.7% of *MT-CYB* sequences for *Capra* and *Ovis* spp. indicates good preservation of ancient mtDNA in the samples, which likely reflects the generally cool continental climate of sampled sites and high-elevation for sites in the IAMC, although there are appreciable differences in the recovery of *MT-CYB* sequences between sites (Table 2). The

**Table 2. Recovery rates of *MT-CYB* sequences by site per taxon.** *MT-CYB* sequences from Begash, Dali, and Tasbas were previously reported [47].

| | | | Frequencies | | | | |
|---|---|---|---|---|---|---|---|
| **Northern Eurasian Steppe** | **Chronology** | **Samples tested** | ***MT-CYB Capra hircus*** | ***MT-CYB Ovis aries*** | ***MT-CYB Capra sibirica*** | ***MT-CYB* Fail** | **Success Rate** |
| **Bozshakol** | Late Bronze Age | 7 | 5 | 1 | 0 | 1 | 85.7% |
| **Air-Tau** | Late Bronze Age | 14 | 3 | 1 | 0 | 10 | 28.6% |
| **Rublevo-VI** | Late Bronze Age | 5 | 2 | 3 | 0 | 0 | 100.0% |
| **Zamiin-Utug** | Iron Age | 2 | 2 | 0 | 0 | 0 | 100.0% |
| Central Kazakhstan | | | | | | | |
| **Taldysai** | Middle Bronze Age | 28 | 4 | 18 | 0 | 6 | 78.6% |
| | Late Bronze Age | 25 | 10 | 9 | 0 | 6 | 76.0% |
| **Myrzhik** | Late Bronze Age | 6 | 2 | 3 | 0 | 1 | 83.3% |
| **Atasu** | Late Bronze Age | 2 | 0 | 0 | 0 | 2 | 0.0% |
| Inner Asian Mountain Corridor | | | | | | | |
| **Begash** | Early Bronze Age | 25 | 2 | 17 | 0 | 6 | 76.0% |
| | Middle Bronze Age | 25 | 2 | 10 | 1 | 12 | 52.0% |
| | Late Bronze Age | 6 | 3 | 0 | 0 | 3 | 50.0% |
| | Iron Age | 10 | 3 | 4 | 0 | 3 | 70.0% |
| | Mongol Period | 1 | 0 | 1 | 0 | 0 | 100.0% |
| | Historical Period | 1 | 0 | 0 | 0 | 1 | 0.0% |
| **Dali** | Early Bronze Age | 27 | 4 | 22 | 0 | 1 | 96.3% |
| | Middle Bronze Age | 5 | 3 | 1 | 1 | 0 | 100.0% |
| | Late Bronze Age | 1 | 0 | 1 | 0 | 0 | 100.0% |
| **Tasbas** | Late Bronze Age | 6 | 4 | 2 | 0 | 0 | 100.0% |
| **Uch-Kurbu** | Late Bronze Age | 2 | 1 | 1 | 0 | 0 | 100.0% |
| **TOTAL** | | **198** | **50** | **94** | **2** | **52** | **73.7%** |

higher frequencies of domesticated sheep versus goat *MT-CYB* sequences generally echo previously published zooarchaeological data for the sampled sites and other sites across the Eurasian steppe zone, showing a ratio of sheep to goat skeletal remains of approximately 10–15:1 [46, 81, 84, 86, 93, 94].

Two specimens from Begash previously identified as Siberian ibex based on morphological criteria [86] were determined to be domesticated sheep, based on *MT-CYB* sequences that are specific to *Ovis aries* [47]. Two additional Siberian ibex specimens did not yield amplicons, likely due to DNA damage resulting from burning that was visible on bone surfaces. One Siberian ibex specimen from Dali dating to the Middle Bronze Age (ca. 1800–1500 cal BC), which was identified based on more reliable cranial morphology, yielded a *Capra sibirica MT-CYB* sequence, while one specimen previously identified as a domesticated caprine from Begash [86] yielded a *Capra sibirica MT-CYB* sequence [47].

Misidentifications of *Capra sibirica* due to shared morphological characteristics with sheep and use of metrical determinations establishing large body-size, a trait typically associated with wild goats, highlights the need to exercise caution when attempting to separate wild and domesticated caprines. Under-identification of Siberian ibex in faunal assemblages limits our understanding of the importance of hunting by ancient pastoralists in Central Asia, which may have been a less common practice than previously suggested [86], and also obscures the significance of large-bodied, male rams in pastoralist livestock herds. In general, morphological criteria used to identify Central Asian sheep and goat draw from those developed for European and Near Eastern populations. Although the skeletal morphology of Siberian ibex is poorly studied, morphological similarity between domesticated sheep and wild ibex in some skeletal elements have been previously identified for both Near Eastern and European populations [95]. While collagen peptide mass fingerprinting (Zooarchaeology by Mass Spectrometry, aka "ZooMS") may be used to differentiate sheep and goat specimens as a relatively cheap biomolecular method [49, 96, 97], this technique is limited to genus-level resolution and, thus, cannot differentiate between wild and domesticated lineages.

## *Capra hircus* haplogroup diversity

Out of 50 samples that yielded *Capra MT-CYB* sequences, we recovered 43 control region sequences. One specimen failed to generate the 193-bp amplicon but yielded a 130-bp sequence of HVR1 using primers CAP-FII and CAP-RII. Although the primers used to amplify sequences of the control region are generally *Capra*-specific, one sample from Taldysai produced a sequence for *Ovis aries*. Control region sequences were confirmed with duplicates taken from independent DNA extractions of each specimen.

Taken together, the ancient sequences of *Capra hircus* represent divergent mtDNA lineages, corresponding to haplogroups A, C, and D (Table 3), which are well characterized based on modern datasets consisting of more than 3000 domesticated and wild goats [15, 17, 18]. In our sample of *Capra hircus* mtDNA sequences, the Late Bronze Age (ca. 1700–1000 BC) is best represented (n = 26), followed by the early-middle Bronze Age (ca. 2700–1700 BC; n = 11) and the Iron Age (ca. 400 BC-AD 200; n = 5).

Diversity of mtDNA haplotypes was lowest for sites located in the NES, indicated by all recovered HVR1 sequences belonging to haplogroup A (Table 3) and an average genetic distance of 0.015±0.005. Low goat mtDNA diversity is in the NES is driven by the high occurrence of one HVR1 haplotype that was shared among three individual goats from Bozshakol and two from Air-Tau. This haplotype is the most common one in the dataset and was also present in five goats from sites located in central Kazakhstan (Fig 2). We are confident that the identical haplotypes from each site represent different individuals based on dental specimens

**Table 3. Recovery of *Capra hircus* mtDNA *MT-CYB* and HVR1 sequences belonging to major haplogroups by site.** One HVR1 sequence of Siberian ibex was also recovered and is not counted here.

| | | Frequencies | | | | | |
|---|---|---|---|---|---|---|---|
| **Northern Eurasian Steppe** | **Chronology** | ***MT-CYB Capra hircus*** | **Haplogroup A** | **Haplogroup C** | **Haplogroup D** | **HVR1 Fail** | **Success Rate** |
| Bozshakol | Late Bronze Age | 5 | 5 | 0 | 0 | 0 | 100.0% |
| Air-Tau | Late Bronze Age | 3 | 3 | 0 | 0 | 0 | 100.0% |
| Rublevo-VI | Late Bronze Age | 2 | 2 | 0 | 0 | 0 | 100.0% |
| Zamiin-Utug | Iron Age | 2 | 2 | 0 | 0 | 0 | 100.0% |
| Central Kazakhstan | | | | | | | |
| Taldysai | Middle Bronze Age | 4 | 3 | 1 | 0 | 0 | 100.0% |
| | Late Bronze Age | 10 | 6 | 1 | 1 | 2 | 80.0% |
| Myrzhik | Late Bronze Age | 2 | 1 | 0 | 0 | 1 | 50.0% |
| Atasu | Late Bronze Age | 0 | 0 | 0 | 0 | 0 | - |
| Inner Asian Mountain Corridor | | | | | | | |
| Begash | Early Bronze Age | 2 | 0 | 1 | 0 | 1 | 50.0% |
| | Middle Bronze Age | 2 | 0 | 0 | 0 | 2 | 0.0% |
| | Late Bronze Age | 3 | 0 | 1 | 1 | 1 | 66.7% |
| | Iron Age | 3 | 1 | 0 | 2 | 0 | 100.0% |
| | Mongol Period | 0 | 0 | 0 | 0 | 0 | - |
| | Historical Period | 0 | 0 | 0 | 0 | 0 | - |
| Dali | Early Bronze Age | 4 | 3 | 0 | 0 | 1 | 75.0% |
| | Middle Bronze Age | 3 | 3 | 0 | 0 | 0 | 100.0% |
| | Late Bronze Age | 0 | 0 | 0 | 0 | 0 | - |
| Tasbas | Late Bronze Age | 4 | 3 | 0 | 1 | 0 | 100.0% |
| Uch-Kurbu | Late Bronze Age | 1 | 0 | 1 | 0 | 0 | 100.0% |
| **TOTAL** | | **50** | **32** | **5** | **5** | **8** | **84.0%** |

being of the same body side or, in cases of specimens representing opposing body sides, individuality is confirmed by age distinction according to tooth wear stages (S1 Table). Interestingly, the remaining sample set from Taldysai is characterized by diverse HVR1 haplotypes, belonging to haplogroups A (n = 4), C (n = 2), and D (n = 1) (Table 3; Fig 2), which together exhibit an average genetic distance of 0.042±0.008.

Mitochondrial diversity was substantially higher for sites located in the IAMC than the NES, shown by sequences belonging to haplogroups A (n = 10), C (n = 3), and D (n = 4) (Table 3; Fig 2), which were unique, except for two C haplotypes from Begash and Uch-Kurbu (Fig 2). Notably, the IAMC is characterized by a high average genetic distance of 0.053±0.010, compared to the NES and central Kazakhstan.

Sequences of domesticated goats recovered from Dali, representing the earliest pastoralist occupations in the study [47], belong to haplogroup A (n = 4), while middle Bronze Age to Iron Age layers of IAMC sites contained an unexpectedly high 3:2 ratio of C and D haplotypes to A haplotypes (Table 3). This haplogroup distribution could indicate an influx of goats with diverse maternal origins into the IAMC after the middle Bronze Age, which strongly contrasts against the predominance of A lineages in post-Neolithic goats across the Near East. Although the sample size for the Iron Age is small, two out of three sequences are of lineage D, suggesting mtDNA diversity may have increased up to the end of the first century BC. This corresponds with the identification of one D lineage goat from Iron Age site of Bancheng in the Inner Mongolia Province of China [61] (Fig 1). Moreover, a recent study documenting mtDNA diversity in modern goats in Kazakhstan found that D lineages only occurred in low

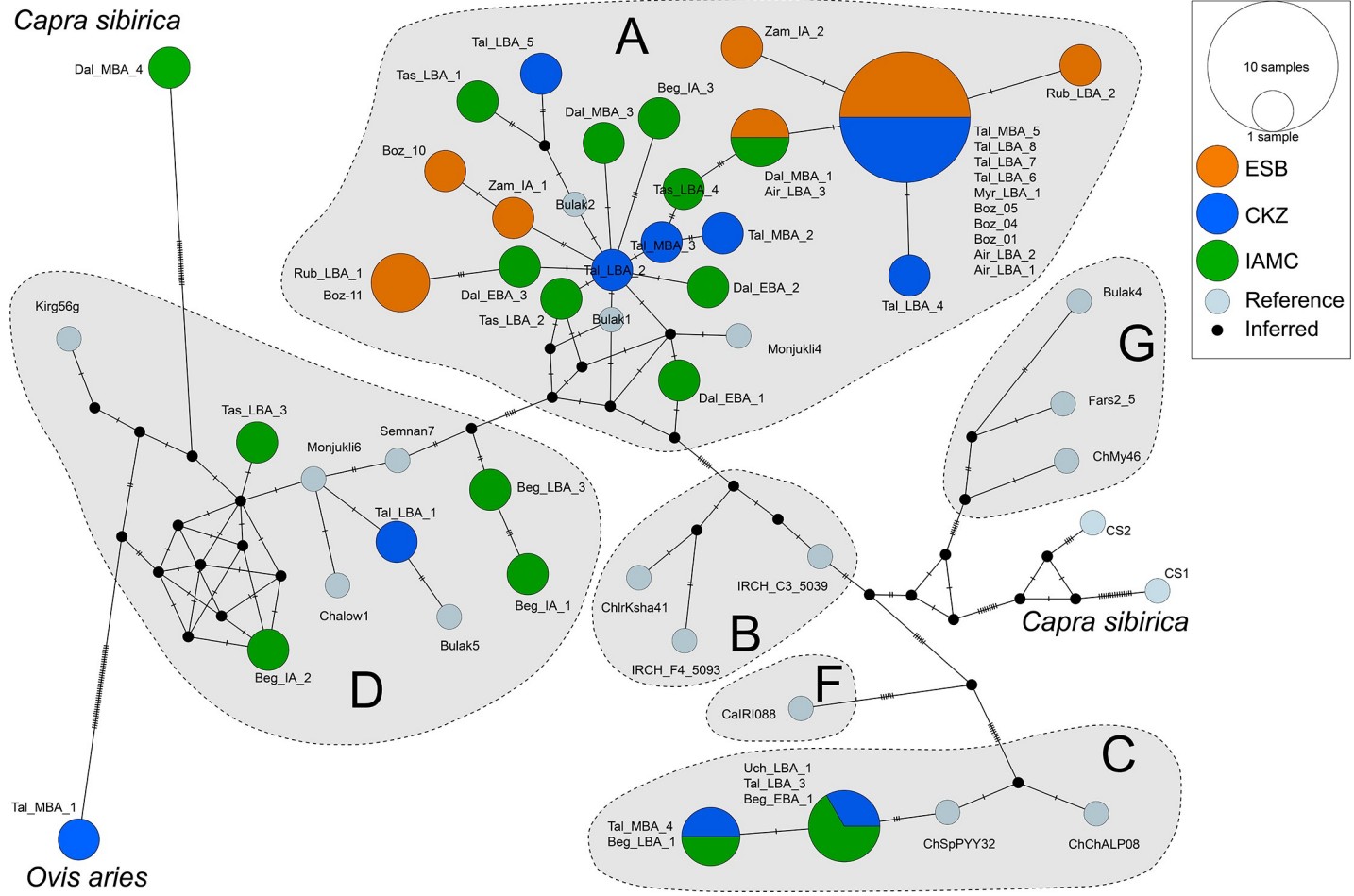

**Fig 2. A minimum spanning haplotype network showing the genetic diversity of domesticated and wild goats analyzed in this study and that represented by reference sequences across the major *Capra hircus* mtDNA haplogroups.**

numbers in the southeastern region of the country compared to other regions that were dominated by A lineages [98]. Taken together, the diversity in mtDNA lineages from these studies and that in our sample reinforce the IAMC as an important locus connecting ancient Central Asian herders to those in east Asia via western China.

Sequences representing haplogroups B, G, or F were not recovered in our study. We did not expect to identify F haplotypes, since goats belonging to the F lineage are the rarest in modern populations, having only been isolated in three Sicilian domesticated goats [99], in addition to wild bezoars from a wide geographic extent spanning the Caucasus and South Asia [15, 16]. Managed goats belonging to haplogroup F (n = 5) were only present in the southern Levant during the Pre-Pottery Neolithic B, while ancient wild bezoars from southern Turkey and Armenia (n = 2) were also identified as belonging to lineage F [14]. We expected to recover B haplotypes from IAMC sites, since they are the second most abundant in modern distributions (ca. 6.5%) [15, 17], and one B haplotype was found among Bronze Age goats in southern Uzbekistan [14] and two B haplotypes were found among Iron Age goats from northern China [61]. Goats of lineage B must have passed through Central Asia as pastoralists extended the social networks through which animals and crops were initially spread across Asia. The absence of G haplotypes in our sample was difficult to predict, since goats belonging

to this lineage were recovered from Neolithic sites in Iran and Turkmenistan [14], which might have been transmitted into Central Asia at similar rates as goats belonging to lineage D. Today, goats of the G lineage are largely restricted to northern and central Africa and have a very low global frequency of 0.9% [15]. The absence of B and G haplotypes in our sample demands further investigation of ancient goat genetic diversity in Iran, the IAMC, and China in order to resolve processes leading to the persistence and extinction of goat lineages in particular times and places.

## Network and phylogenetic analysis

Ancient goat sequences recovered in this study are scattered throughout the A clade, which is likely an outcome of A haplotypes being the most dominant in domesticated goats after large population expansions of A groups since domestication [14, 16, 17, 19]. Haplotypes of the C lineage from Begash, Uch-Kurbu, and Taldysai cluster together and are distinct from modern reference haplotypes (Fig 2). Unexpectedly, two out of the four D haplotypes from goats in the IAMC during the Bronze and Iron Ages are newly discovered D haplotypes (Fig 2; Beg_LBA_3 and Beg_IA_1), which cluster phylogenetically as a distinct sub-clade within this haplogroup (Fig 3). This finding suggests the possibility of genetic continuity of rare domesticated goat haplotypes at Begash over a long period of time from 1600 to 20 cal BC, which spans prominent cultural turnovers during the Bronze to Iron Ages. Interestingly, no modern descendant or closely related sequences of this D lineage are known to exist. Haplotypes of lineage D are rare in contemporary world-wide distributions (0.6%) [15, 17], but are more common in bezoar populations on the Iranian plateau (ca. 13%) [15]. Haplotypes of the D lineage have also been recovered from Neolithic and Bronze Age sites in Israel, Iran, Turkmenistan, and Uzbekistan [14], which cluster with the other two Bronze and Iron Age haplotypes from the IAMC on the main D clade (Fig 2). Given our recovery of a relatively high frequency of C and D haplotypes in the IAMC, ancient frequencies of these lineages were likely far higher in ancient Central Asia than in modern distributions. The presence of these lineages further indicates high mtDNA diversity of goats in the mountain regions of Central Asia compared to the NES that is suggestive of different intensities and geographic extents of social interaction between and among these regions.

Due to random genetic drift ultimately leading to monomorphism, a single lineage may become dominant in a diverse population after many generations in a population that has little inward gene flow of assorted lineages [100, 101]. While it is impossible to estimate the precise population size of any particular livestock species based on zooarchaeological data, goat skeletal remains are about 10–15 times less frequent than those of sheep and about five times as infrequent as cattle remains from sites in the IAMC during the Bronze and Iron Ages [46, 86, 93], indicating that goat population sizes were relatively small compared to other domesticated species. Without a large population to buffer the effects of genetic drift, the persistence of C and D lineages in goats is likely due to high-rates of gene flow along the IAMC, likely connecting goat herds in the IAMC with diverse goat populations in southern Central Asia or further southwest on the Iranian Plateau at consistent rates. Frequent exchange events over relatively short distances could have linked goat herds of pastoralist communities as a large, global effective population, thus maintaining the prevalence of less common mtDNA lineages compared to the NES. A high degree of community interaction, likely driven by both seasonal mobility of pastoralists [20, 45, 102] and sprawling economic networks for the mining and exchange of copper and tin to produce bronzes during the Bronze Age and Iron Age [36, 103–105], could have been a main factor for promoting inter-regional interactions that went hand in hand with gene flows among livestock herds at inter-regional scales.

In contrast, the NES was characterized by goats exhibiting A haplotypes with low genetic diversity, as indicated by identical sequences among NES sites and a higher connectivity within the haplotype network (Fig 2). This pattern could have been the result of domesticated goats spreading eastward across the NES from a monomorphic founding population of goats, which later during the Bronze and Iron Ages was not characterized by inward gene flows of goats exhibiting C or D haplotypes that were present in the IAMC. While the NES has long been conceived as an important ecozone of pastoralist mobility and interaction, our observation of low goat mtDNA variation in this region suggests that, at least in part, there were limited communication and exchange with southern regions in Central Asia, especially the IAMC, where rare goat haplotypes and higher mtDNA diversity persisted in the Bronze and Iron Ages.

Interestingly, Taldysai located in central Kazakhstan was characterized by an intermediate level of goat mtDNA diversity relative to the NES and IAMC (Fig 2). This finding provides an important reconfiguration in how traditional cultural interaction spheres of the Eurasian steppe zone are defined, suggesting that central Kazakhstan served as an arena of interaction between both north and south regions but did not function as a corridor for goat gene flow connecting the north and south together. This finding corresponds to recent research on regional variance in domesticated sheep astragali morphology using geometric morphometrics, showing that localized populations likely were genetically isolated from one another [106]. However, the presence of C and D goat haplotypes at Taldysai suggests repeated exchange events with southern Central Asia and the IAMC, but these lineages likely did not pass along to NES sites. Human paleogenetic studies have described a substantial amount of human gene flow from pastoralist communities in the western NES to southern Central Asia during the second millennium BC [107, 108], which would have most certainly been accompanied by goats predominantly characterized by A haplotypes. Similarly, the recent discovery by Narasimhan et al. [31] of limited human gene flows from Turan and the IAMC into central Kazakhstan during this period would explain the presence of some goat lineages at Taldysai that were recovered in high frequencies at IAMC sites.

The one ancient *Capra sibirica* sequence recovered from Dali phylogenetically clustered as a sister group to previously published modern *Capra sibirica* sequences, which is supported by a 99% Mr. Bayes posterior probability (Fig 3). However, the recovered *MT-CYB* and HVR1 sequences from this *Capra sibirica* specimen did not cluster with the *Capra sibirica* reference sequences in the haplotype network (Fig 2), which is likely due to highly divergent mitochondrial variation between ancient and modern populations; the former probably had far greater genetic variation than the latter. To date, our *Capra sibirica* sequences represent the only ancient genetic sequences recovered from *Capra sibirica* and demonstrates a newly discovered degree of diversity in this species, while so far being poorly characterized genetically and biogeographically [109–113].

## Conclusions

Despite small sample sizes, the results of this study suggest higher levels of gene flow among domesticated goats along the IAMC, which is likely a reflection of increased community interaction in the foothill zones, compared to the open steppe landscape. The northern "Northern Eurasian Steppe" is typically viewed as a metaphorical highway of cultural interaction between eastern Europe and Siberia, since the initial spread of pastoralism to the Altai Mountains via purported migrations of Afanasievo communities [26] and through the formation of expansive cultural horizons of the Iron Age [39–44]. Based on the low mtDNA diversity present in Bronze Age and later Iron Age domesticated goats in the NES, there may be an unexpectedly low degree of admixture between northern steppe pastoralist communities and those in

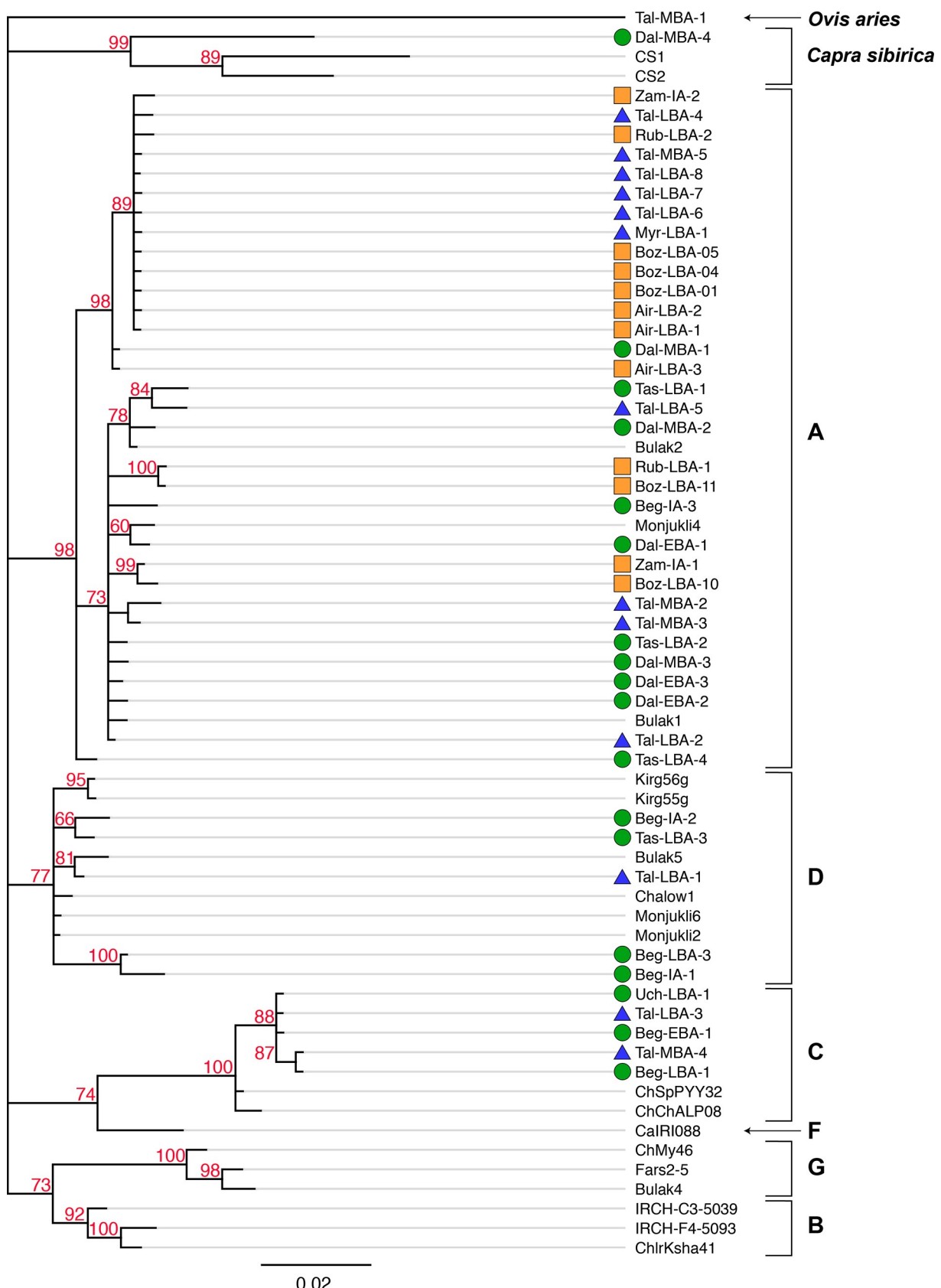

**Fig 3. Phylogenetic tree using MrBayes 3 of ancient mtDNA sequences (HVR1 and MT-CYB regions) identified in this study (colorful symbols reflect site groups show in Fig 1), in addition to previously published ancient and contemporary reference sequences from Eurasia;** *Ovis aries* **was used as the outgroup.** Posterior probabilities ≥ 60% are shown on each branch. Haplogroup designations of *Capra hircus* are listed to the right of the sample ID numbers.

adjacent regions. The goats analyzed from sites in the NES appear to be consistent with a founder population that was likely present in eastern Europe after an earlier arrival of domesticated goats characterized by A haplotypes from the Near East via Eastern Europe or the Caucasus.

In the IAMC, we observed a possible shift in the frequencies of mtDNA lineages from the early-middle Bronze Age (2700–1600 cal BC) to the late Bronze Age and Iron Age (1600–20 cal BC), which may be a result of reconfigured socio-economic networks that facilitated transfers of goats belonging to C and D lineages to the region from diverse genetic pools from agro-pastoralist communities located in the Iranian plateau. In the absence of individual animals moving long distances along the IAMC, genetic isolation between discrete goat populations may have been greatly diminished due to directed breeding patterns of goats by pastoralists that effectively created a macro-population of goats, thus maintaining diverse maternal lineages. These findings provide additional support that the IAMC was a dynamic arena of cultural activity spanning distant regions of Central and Inner Asia, which ultimately linked these regions to contemporaneous cultural arenas in east Asia [20]. From Taldysai, the recovery of C and D haplotypes and several A haplotypes that were identical to those from NES sites suggests the region of central Kazakhstan was discretely connected to interaction spheres of communities in the NES and IAMC but may have marked a boundary between them.

More research is needed to resolve the precise nature of livestock translocations, inter-regional genetic networks, and selective pressures on livestock. This study offers a much-needed expansion of datasets for characterizing ancient goat mitochondrial diversity in Eurasia, and larger sample sizes spanning more sites along proposed transmission routes will improve the quality of inferences concerning inter-regional human interactions that can be drawn from maternal lineages. The addition of other domesticated taxa, such as sheep and cattle, to future paleogenetic analysis will help establish differences in genetic sourcing from various geographical regions. By also examining genetic variability in wild taxa, ancient biogeographic patterns may be identified to reconstruct a more complete picture of human engagements with the landscape that may reveal critical patterns in the relationship between hunting and herding strategies. The alignment of new evidence for these subsistence pursuits with recent findings of early and intensive cultivation of domesticated crops by Central Asian communities during the Bronze Age will further clarify the role that economic diversification played in shaping regional and trans-regional social interaction spheres.

## Supporting information

**S1 Text. Descriptions of archaeological sites and references.**
(DOCX)

**S1 Table. List of aDNA samples, taxonomic identifications, haplogroup designations, skeletal element identifications, tooth wear stages, archaeological context information, and archival location.** Mandibular wear stages follow Payne [65]. Maxillary wear stages are based on a relative system developed by zooarchaeologist Sarah Pleuger: scale 0–5, denoted with symbol *.
(XLSX)

**S2 Table. PCR master mix formula.**
(XLSX)

**S3 Table. Primer sequences and annealing temperatures.**
(XLSX)

**S4 Table. Reference sequences used for phylogenetic and network analysis.**
(XLSX)

**S1 File. FASTA file containing *MT-CYB* and HVR1 sequences from samples tested and references.**
(FASTA)

**S1 Fig.**
(TIF)

## Acknowledgments

We thank the Institute of Clinical Molecular Biology in Kiel for providing Sanger sequencing as supported by the DFG Clusters of Excellence "Precision Medicine in Chronic Inflammation" and "ROOTS." We are grateful to Tanja Naujoks for technical support. We also thank Lisa Böhme and Yekaterina Dubyagina for valuable assistance during the course of this research. We acknowledge financial support by DFG within the funding programme Open Access Publizieren.

## Author Contributions

**Conceptualization:** Taylor R. Hermes.

**Formal analysis:** Taylor R. Hermes.

**Funding acquisition:** Cheryl A. Makarewicz.

**Investigation:** Taylor R. Hermes.

**Methodology:** Taylor R. Hermes.

**Resources:** Taylor R. Hermes, Michael D. Frachetti, Dmitriy Voyakin, Antonina S. Yerlomaeva, Arman Z. Beisenov, Paula N. Doumani Dupuy, Dmitry V. Papin, Giedre Motuzaite Matuzeviciute, Jamsranjav Bayarsaikhan, Jean-Luc Houle, Alexey A. Tishkin, Almut Nebel, Ben Krause-Kyora.

**Supervision:** Almut Nebel, Ben Krause-Kyora, Cheryl A. Makarewicz.

**Validation:** Taylor R. Hermes.

**Visualization:** Taylor R. Hermes.

**Writing – original draft:** Taylor R. Hermes.

**Writing – review & editing:** Taylor R. Hermes, Michael D. Frachetti, Dmitriy Voyakin, Antonina S. Yerlomaeva, Arman Z. Beisenov, Paula N. Doumani Dupuy, Dmitry V. Papin, Giedre Motuzaite Matuzeviciute, Jamsranjav Bayarsaikhan, Jean-Luc Houle, Alexey A. Tishkin, Almut Nebel, Ben Krause-Kyora, Cheryl A. Makarewicz.

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
