## [Decision Letter · Decision Letter 0]

10 Dec 2019

PONE-D-19-29815

High mitochondrial diversity of domesticated goats persisted among Bronze and Iron Age pastoralists in the Inner Asian Mountain Corridor

PLOS ONE

Dear Dr. Hermes,

Thank you for submitting your manuscript to PLOS ONE. After careful consideration, we feel that it has merit but does not fully meet PLOS ONE’s publication criteria as it currently stands. Therefore, we invite you to submit a revised version of the manuscript that addresses the points raised during the review process.

all comments need to be addressed.

We would appreciate receiving your revised manuscript by Jan 24 2020 11:59PM. To enhance the reproducibility of your results, we recommend that if applicable you deposit your laboratory protocols in protocols.io, where a protocol can be assigned its own identifier (DOI) such that it can be cited independently in the future. For instructions see: http://journals.plos.org/plosone/s/submission-guidelines#loc-laboratory-protocols

We look forward to receiving your revised manuscript.

Kind regards,

Peter F. Biehl, PhD

Academic Editor

PLOS ONE

Journal Requirements:

2) In your manuscript, please provide additional information regarding the specimens used in your study. Ensure that you have reported specimen numbers and complete repository information, including museum name and geographic location. If permits were required, please ensure that you have provided details for all permits that were obtained, including the full name of the issuing authority. For more information on PLOS ONE's requirements for paleontology and archaeology research, see https://journals.plos.org/plosone/s/submission-guidelines#loc-paleontology-and-archaeology-research.

3) We note that Figure 1 in your submission contains map/satellite images which may be copyrighted. All PLOS content is published under the Creative Commons Attribution License (CC BY 4.0), which means that the manuscript, images, and Supporting Information files will be freely available online, and any third party is permitted to access, download, copy, distribute, and use these materials in any way, even commercially, with proper attribution. For these reasons, we cannot publish previously copyrighted maps or satellite images created using proprietary data, such as Google software (Google Maps, Street View, and Earth). For more information, see our copyright guidelines: http://journals.plos.org/plosone/s/licenses-and-copyright.

You may seek permission from the original copyright holder of Figure(s) [#] to publish the content specifically under the CC BY 4.0 license. 

If you are unable to obtain permission from the original copyright holder to publish these figures under the CC BY 4.0 license or if the copyright holder’s requirements are incompatible with the CC BY 4.0 license, please either i) remove the figure or ii) supply a replacement figure that complies with the CC BY 4.0 license. Please check copyright information on all replacement figures and update the figure caption with source information. If applicable, please specify in the figure caption text when a figure is similar but not identical to the original image and is therefore for illustrative purposes only.

4) Thank you for stating the following in the Competing Interests section:

"The authors have declared that no competing interests exist.".

We note that one or more of the authors are employed by a commercial company:Archaelogical Expertise, LLC, Almaty, Kazakhstan.

5) Please include captions for your Supporting Information files at the end of your manuscript, and update any in-text citations to match accordingly. Please see our Supporting Information guidelines for more information: http://journals.plos.org/plosone/s/supporting-information.

Additional Editor Comments (if provided):

Your manuscript has now been seen by two referees, whose comments are appended below. You will see from these comments that the referees find your work of high interest. In light of these comments, only minor revisions are needed - see especially reviewer 2 with the suggestion to move the site description to the supplement.

Reviewers' comments:

Reviewer's Responses to Questions

**Comments to the Author**

1. Is the manuscript technically sound, and do the data support the conclusions?

Reviewer #1: Yes

Reviewer #2: Yes

2. Has the statistical analysis been performed appropriately and rigorously? 

Reviewer #1: Yes

Reviewer #2: Yes

3. Have the authors made all data underlying the findings in their manuscript fully available?

Reviewer #1: Yes

Reviewer #2: Yes

4. Is the manuscript presented in an intelligible fashion and written in standard English?

Reviewer #1: Yes

Reviewer #2: Yes

5. Review Comments to the Author

Reviewer #1: This manuscript is extremely well written in a very clear and well-organized style. It will require very little correcting, as I failed to find any typos or grammatical errors. The authors are highly qualified, with extensive years of experience in Asian steppe archaeology, focused on the Bronze and Iron Age. The ancient mtDNA, stable isotope and cementum analysis, as well as standard morphological analysis of plant and animal remains form a solid, holistic picture of goat herding in the Djungar Mountains and surrounding regions of Siberia, northern and central Kazakhstan and northwestern Mongolia. The sites from which samples were derived were selected in a meaningful way to capture the evidence to develop their results on subsistence and caprine breeding and maintenance. The sample sizes were ample, although future expansion of data is certainly welcome. Here I would like to briefly summarize their most significant results.

The found that diversity of mtDNA haplotypes was lower for NES sites and higher for IAMC sites, which has major implications for these two regions. They interpret some of the genetic findings as an indication of connections with western China. The absence of B and G haplotypes is interesting and shows the need to increase sample sizes from Iran and China before the migrations of goat populations through Central Asia can be better understood.

One interesting and potentially very significant finding was that morphological identification of wild species of caprines conflicts with their ancient mtDNA results. This casts doubt on previous publications that report hunting of wild species when, in fact, the remains may just be from large males in the domestic livestock. This would mean that hunting was less significant than originally thought, at least for caprines, and that size variation within domestic herds is something worth considering. It could imply that selection for particular characteristics and perhaps even breed development was being initiated in these livestock populations or, as they indicate, the influx of goats from outside after the middle Bronze Age.

Another important development is the assessment of the low frequencies of goats vs. sheep and cattle. This can shed light on the importance of their uses. Goats tend to be bigger milk producers than sheep, and, though the hair of certain breeds is highly valued, sheep would yield wool. The amount of meat they produce is basically the same, so perhaps these populations depended more on cow milk than goat and considered wool production more important than goat hair (if the goats had cashmere or angora-type coats, which is unknown at present-they may have had a short coat).

This paper sheds light on the degree of human population migration owing to pastoral mobility and the bronze trade in the Bronze and Iron Ages. For the NES, there seems to have been less mobility and exchange than in the IAMC. This research marks a significant beginning in understanding connections between human and livestock populations, most notably the activities in Central Kazakhstan as a possible intermediate region, but also a border between the north and south. They interpret the low diversity in the NEC as reflecting the eastward migration of Near Eastern goats from Eastern Europe via the Afanasievo culture (and perhaps others) to the Altai region. The IAMC, on the other hand, was more dynamic, with connections potentially with Iran and China. This exciting research is definitely a great start, but, as they point out, their tantalizing results show that much more work needs to be done in the future to fine-tune our understanding of mobility of herds and people in Central Asia and connections between Eastern Europe, the Near East, and the far East. I look forward to their continuing research and new results and hope that they will also be able to address issues such as breed development and product exploitation (meat, coat, and dairy) through genetic and traditional archaeological information.

Reviewer #2: I think this paper presents novel and interesting data and is professionally prepared and presented. I suggest a few minor changes and reorganizations but otherwise supports its acceptance for publication.

In this paper, the authors present new mt DNA sequences for archaeological (Bronze and Iron Age) goats from two regions 'Central Asia' including the North Eurasian Steppe (NAS) and the Inner Asian Mountain Corridor (IAMC). The authors present mt sequences and haplogroup assignments for 50 domestic goat specimens from 10 sites. This itself is a valuable contribution to the understanding of the historical genetics of goat populations as well as the migration and interaction histories of these two regions. Even more interesting is the patterns the authors identify in the mt data. They point out that goats from the NAS region have low mt diversity, all characterized as mt haplogroup A. In contrast, the goats from the IAMC region reflect higher variation with low frequencies of haplogroup A while also exhibiting haplogroups C and D. From these patterns the authors link the IAMC region with goat diversity on the Iranian plateau whereas the A lineages in the NAS perhaps derive from eastern Europe or the Caucasus region and thus reflect divergent (goat) colonization histories.

These are very interesting data and patterns that fit into the very high profile current discussions of the role of mobility, trade and exchange, and population movements in and through these central Asian corridor regions.

My main concern with reading the paper comes from the organization. I feel like the section describing the sites is intrusive and disrupts the narrative. I suggest moving the site descriptions to a supplement rather than including them in the main text of the paper. I also think it should be easier to identify which specimens represent which haplogroup--this should be included in the main specimen table in Supplement table 1.

Specific and generally minor comments/questions/suggestions:

Line 85: what do the authors mean by saying that mobile pastoralism emerged in the 5th millennium BC? What does ‘mobile’ mean?

Line 102: descriptions of the routes for the spread of domestic animals make it sound like a simple process of migration. Does not address the fact that these regions were occupied by local hunter-gatherers and there are active discussion concerning the mechanisms by which domesticates moved into this broad region.

Line 196-330: this section describing the sites is usually moved to a supplemental file in aDNA publications. I would recommend moving this text and adding a Table to the main text with the site names, dates and perhaps a very brief description of the most relevant details.

Line 365: I am not sure that this sentence is relevant here since it is not referring specifically to

Capra sibirica which, based on Pidancier et al 2006, seems to be a more basal member of the Capra genus, although Sardina et al 2006’s analysis shows Capra sibirica closely related to bezoar and domestic goat lineages. Either way, it is not clear that this comment is relevant unless it is specifically directed at Capra sibirica morphology rather than ‘ibexes’ more generally.

Line 420: lineage D is described as ‘rare’. It is rare based on a study of modern goats in Naderi et al 2007 but based on Daly et al 2018, lineage D is not all that rare—it is present in Neolithic and Bronze Age Iran as well as Chalcolithic Israel. Based on ancient samples in Daly et al 2018 and Colli et al 2015, C is much more rare and is found in wild Iranian bezoar as well as Neolithic domestic goats in the Balkans, modern domestic goats in central Europe, and modern domestic goats in Iran. As a result, C is a less good indicator of the migration direction.

Line 431: Distribution of F is greater than stated here. Based on Colli et al 2015, haplogroup F goats have also been found in wild bezoar in Iran. The authors also report 1 modern Iranian domestic goat exhibiting the F haplogroup (Supplement Table 4). Colli et al 2015 also report that F is also present in wild bezoar from the Caucasus to Pakistan which I think is referencing data from Naderi et al 2008 also report that F is present in modern bezoar populations in southern Turkey and eastern Turkey as well.

Line 475: change wording of “more infrequent” to less frequent

Line 477: this statement is especially true since goats have higher reproductive potential than both sheep and cattle.

Line 521: reword “…while for now…”

Why are the mt haplogroups not labeled in the list of specimens in the Supplemental Tables?

Figure 2: the CKZ label is not identified on the map in Figure 1.

Figure 2: why do the 2 Capra sibirica samples not cluster together? Does this suggest hybridizing with domestic goat populations or just divergent wild Capra mt haplogroups?

6. PLOS authors have the option to publish the peer review history of their article (what does this mean?). If published, this will include your full peer review and any attached files.

Reviewer #1: Yes: Sandra L. Olsen

Reviewer #2: No

---

## [Author Response · Author response to Decision Letter 0]

11 Feb 2020

Thank you for the opportunity to revise our manuscript “High mitochondrial diversity of domesticated goats persisted among Bronze and Iron Age pastoralists in the Inner Asian Mountain Corridor.” I would also like to thank the two reviewers for carefully reading this work and providing constructive criticism. I have applied all the changes suggested by you and reviewer #2, and below I will briefly summarize these modifications to the manuscript. (Reviewer #1 did not request corrections.)

I made changes to the formatting of the manuscript to conform to the journal’s style requirements, including the naming of attached files for figures and supporting information. I reviewed S1 Table and filled in missing specimen information, such as skeletal element identifications and archival location. Regarding Figure 1, which was flagged for displaying map imagery, I would like to point out that this imagery is in the public domain provided by naturalearthdata.org. I added a note about its origin and public domain status to the caption of Figure 1. For the supporting information materials, I added captions at the end of the manuscript that describe their contents.

In response to reviewer #2, I applied all suggested changes. Most substantively, I moved the descriptions of the archaeological sites to the supplementary materials (S1 Text) and replaced these with a concise table that includes basic information about the sites and in-text citations. The reviewer aptly noticed that the haplogroup designations of domesticated goats were not listed in S1 Table, and I have added these.

Reviewer #2’s suggestion to define mobile pastoralism in the introduction of the manuscript was good, and I have added a clause to the sentence describing the emergence of this lifeway in the Near East.

Reviewer #2’s idea to modify the sentence in the introduction about the spread of pastoralists into Inner Asia in order to reflect this phenomenon as a process rather than a homogenous event was on point. I made a simple change here to describe this as the spread of “pastoralist subsistence”, which could be either people moving or cultural transmissions of subsistence technologies.

Reviewer #2 noticed that we used a citation for the confusing skeletal morphology between ibex and domesticated sheep and goat that targets European and Near Eastern populations. This was intended to cover Siberian ibex. I agree that this was an improper citation, so I corrected the sentence to reflect a likelihood that Siberian ibex may also be confused with domesticated caprines, given what we know about these scenarios for European and Near Eastern populations.

Reviewer #2’s suggestion to move away from using “rare” to describe D lineage haplotypes of domesticated goats was good. I corrected this throughout the text, in addition to noting that C haplotypes are also notably rare in ancient distributions. Along these lines, I modified the text that describes F haplotypes in domesticated goats and ibexes following Reviewer #2’s suggestions.

Reviewer #2 kindly gave very minor suggestions for wording of a couple phrases , which I changed accordingly. I also corrected Figure 1 to display the “Central Kazakhstan” label for this analytical group. 

Lastly, Reviewer #2 noticed that our ancient Capra sibirica sequences did not cluster with modern Capra sibirica sequences in the haplotype network shown in Figure 2. I added text to the section describing this species in the manuscript that suggests the disagreement is the result of divergence between modern and ancient populations. Hybridization between Siberian ibex and domesticated goats would not explain this observation, since mitochondrial DNA is inherited as a whole unit. 

I would like to sincerely thank you and the reviewers for considering this manuscript for publication. I look forward to hearing from you soon.

---

## [Editor Report · Decision Letter 1]

5 May 2020

High mitochondrial diversity of domesticated goats persisted among Bronze and Iron Age pastoralists in the Inner Asian Mountain Corridor

PONE-D-19-29815R1

Dear Dr. Hermes,

We are pleased to inform you that your manuscript has been judged scientifically suitable for publication and will be formally accepted for publication once it complies with all outstanding technical requirements.

With kind regards,

Peter F. Biehl, PhD

Academic Editor

PLOS ONE
---

## [Editor Report · Acceptance letter]

8 May 2020

PONE-D-19-29815R1 

High mitochondrial diversity of domesticated goats persisted among Bronze and Iron Age pastoralists in the Inner Asian Mountain Corridor 

Dear Dr. Hermes:

I am pleased to inform you that your manuscript has been deemed suitable for publication in PLOS ONE. Congratulations! Your manuscript is now with our production department. 

With kind regards,

on behalf of

Dr. Peter F. Biehl 

Academic Editor

PLOS ONE